# Chromosome drives via CRISPR-Cas9 in yeast

Hui Xu[1,2], Mingzhe Han [1,2], Shiyi Zhou[1,2], Bing-Zhi Li[1,2], Yi Wu [1,2✉] & Ying-Jin Yuan [1,2]

Self-propagating drive systems are capable of causing non-Mendelian inheritance. Here, we report a drive system in yeast referred to as a chromosome drive that eliminates the target chromosome via CRISPR-Cas9, enabling the transmission of the desired chromosome. Our results show that the entire *Saccharomyces cerevisiae* chromosome can be eliminated efficiently through only one double-strand break around the centromere via CRISPR-Cas9. As a proof-of-concept experiment of this CRISPR-Cas9 chromosome drive system, the synthetic yeast chromosome X is completely eliminated, and the counterpart wild-type chromosome X harboring a green fluorescent protein gene or the components of a synthetic violacein pathway are duplicated by sexual reproduction. We also demonstrate the use of chromosome drive to preferentially transmit complex genetic traits in yeast. Chromosome drive enables entire chromosome elimination and biased inheritance on a chromosomal scale, facilitating genomic engineering and chromosome-scale genetic mapping, and extending applications of self-propagating drives.

[1] Frontier Science Center for Synthetic Biology and Key Laboratory of Systems Bioengineering (Ministry of Education), School of Chemical Engineering and Technology, Tianjin University, 300072 Tianjin, China. [2] Collaborative Innovation Center of Chemical Science and Engineering (Tianjin), Tianjin University, 300072 Tianjin, China. ✉email: yi.wu@tju.edu.cn

Biased inheritance is a unique phenomenon that allows self-propagated genetic elements to be transmitted to their progenies with higher frequency than that of normal Mendelian inheritance. Selfish genetic elements such as transposons and homing endonucleases, which have control over their own transmission, can increase their own abundance in species[1–7]. Gene drive is a well-known system utilizing homing endonucleases or CRISPR–Cas9 for generating biased inheritance among population[8]. This system has shown a great capability to transmit desired genes in yeast, insects, and mammals[9–16]. In yeast, a CRISPR-based gene drive was used to verify the concept, efficacy and safeguards of driven genetic alteration by sexual reproduction[10, 11]. In addition, by targeting the intron 4-exon 5 boundary of *Anopheles gambiae* in a caged experiment, a gene drive achieved 100% invasion in mosquitoes within 7–11 generations, leading to the sterility of females to prevent the propagation of malaria[12]. Recently, gene drives mediated by CRISPR–Cas9 were also demonstrated in the female germline of mice[16]. In addition, other drive systems such as toxin-antidote CRISPR gene drive system have been designed to transmit target genetic elements by reducing the fitness of progeny without drive elements[17–19].

Here, we develop a strategy of meiotic drive, called a "chromosome drive", that utilizes chromosomes as drive cassettes to realize biased inheritance of large-scale genetic segments in yeast. We find that specific cleavage via CRISPR–Cas9 near the centromere of a *Saccharomyces cerevisiae* (*S. cerevisiae*) chromosome can cause the loss of the entire chromosome and enable chromosome drive. As proof of concept, chromosome X (chrX), which encodes a green fluorescent protein (GFP) or the multigenic violacein pathway, is transmitted among synthetic yeasts via chromosome drive. We also demonstrate the applications of chromosome drives for transmitting specific chromosomal structures and complex genetic traits.

## Results

**Chromosome elimination by CRISPR–Cas9 in *S. cerevisiae*.**
Centromeres are responsible for chromosome segregation during cell division and are vital for maintaining the stability of chromosomes[20–22]. In *S. cerevisiae*, a point centromere can host an entire chromosome up to the length of several megabase pairs or even an entire genome[20,23]. In this study, we found that an entire yeast chromosome was eliminated by only one double-strand break (DSB) around the centromere via CRISPR–Cas9. In a haploid strain, elimination of any chromosome will make yeast inviable[24]. Thus, all the chromosome elimination experiments in this study were performed on diploid strains. Within synthetic yeast chromosome III (synIII), 186 pairs of dispersed gene-associated PCRTags could be used as a DNA watermarking system to distinguish synthetic yeast chromosomes from wild-type chromosomes (Supplementary Fig. 1)[25–27]. Thus, the PCRTag-labeled synthetic yeast chromosome is an excellent model for characterizing the elimination of the entire chromosome.

For biosafety and efficiency, a molecular containment strategy was applied[10]. We designed a guide RNA (gRNA) targeting a specific sequence (5′-CATTATACGAAGTTATAAGTTGG-3′) that is 135 bp from the centromere of synIII, which does not exist in chromosome III (chrIII) of BY4741 or in wild-type yeast. The CRISPR–Cas9-induced DSB was processed by mating haploid strain BY4741 and synIII strain, harboring Cas9 and gRNA plasmids, respectively (Fig. 1a). The results of a PCRTag analysis showed that all 186 synthetic PCRTags of synIII were lost in the tested diploid strain (yHX0295), indicating that the entire synIII was knocked out. All 186 wild-type PCRTags were present in yHX0295, revealing that no homologous recombination (HR)

occurred between synIII and chrIII during the chromosome elimination (Fig. 1b; Supplementary Fig. 2). In addition, whole-genome sequencing and Southern blot analysis of haploid control strains and yHX0295 also showed that the entire synIII was eliminated in the CRISPR-containing diploid strains (Fig. 1c and Supplementary Figs. 3 and 4). The mating type of *S. cerevisiae* was determined by the expression of *MAT*a or *MAT*α at the *MAT* locus on the sex chromosome synIII. The elimination of synIII in the diploid strain will result in the deletion of *MAT*α at the *MAT* locus, leading to a mating type switch from Aα to A. To characterize the elimination of synIII further, the mating type of the CRISPR-containing diploid strains was tested by tester A strain and tester α strain. The result showed that 78.78% of CRISPR-containing diploid strains could mate again with tester α but could not mate with tester A, confirming the elimination of synIII and the capability of a change in mating type in diploid strains by the use of this method (Fig. 1d and Supplementary Table 1).

To assess the relationship between chromosome elimination and the position of DSBs, seven positions on the left arm of synthetic chromosome X (synX) were chosen to be cut, and elimination of synX was characterized for each position. A synthetic CRISPR-targeted sequence (5′-GTTGCAAATGCTCC GTCGAC**GGG**-3′) was inserted 50, 100, 150, 200 bp, 1, 208, (middle of the left arm of synX) and 415 kb (end of the left arm of synX) from the centromere of synX. Each modified synX harboring episomal gRNA plasmids was mated with BY4742 harboring episomal Cas9 plasmids. For each position, a total of 360 colonies were analyzed via PCRTagging. The results showed that chromosome-elimination efficiency surpassed 80% when the specific cleavage occurred within 200 bp from the centromere of synX, while there was no chromosome elimination when the specific cleavage occurred 1 kb from the centromere of synX (Fig. 1e and Supplementary Table 2). These results indicated that the entire chromosome could be efficiently eliminated through only one DSB near the centromere via CRISPR-Cas9 in *S. cerevisiae*.

**Chromosome drives with a synthetic metabolic pathway.** Yeast is a model organism whose sexual reproduction can be manually controlled in the laboratory; thus, it is suitable for safeguarding experiments involving self-propagating drive systems[10]. Yeast chromosomes can be individually lost in diploid strains by adding a *GAL1* promoter in *cis* to centromere sequences, and most *S. cerevisiae* 2n−1 strains can spontaneously endoreduplicate to 2n strains[28]. Therefore, we designed a proof-of-concept experiment of CRISPR–Cas9 chromosome drive to eliminate synX (harboring a red fluorescent protein (RFP) gene) and duplicate its counterpart wild-type chrX (harboring a GFP gene) via sexual reproduction. Compared with normal inheritance, the chromosome drive system will result in biased inheritance of the desired chromosome and will yield four green spores within one tetrad. In our experiment, the synthetic CRISPR-targeted sequence (5′-GTTGCAAATGCTCCGTCGAC**GGG**-3′) was inserted 50 bp from the centromere on the left arm of synX, and the RFP gene was inserted 156 kb from the centromere on the right arm of synX in strain yHX0299, while the GFP gene was inserted 320 kb from the centromere on the left arm of chrX in strain yHX0141. The two strains described above were mated to initiate the process of chromosome elimination. After sporulation of the mated diploid strains, the fluorescence of the tetrads was observed via a fluorescence microscope. The results showed that all four spores of the observed tetrads displayed green fluorescence and no red fluorescence, while the control group of undriven tetrads displayed two green spores and two red spores. These results

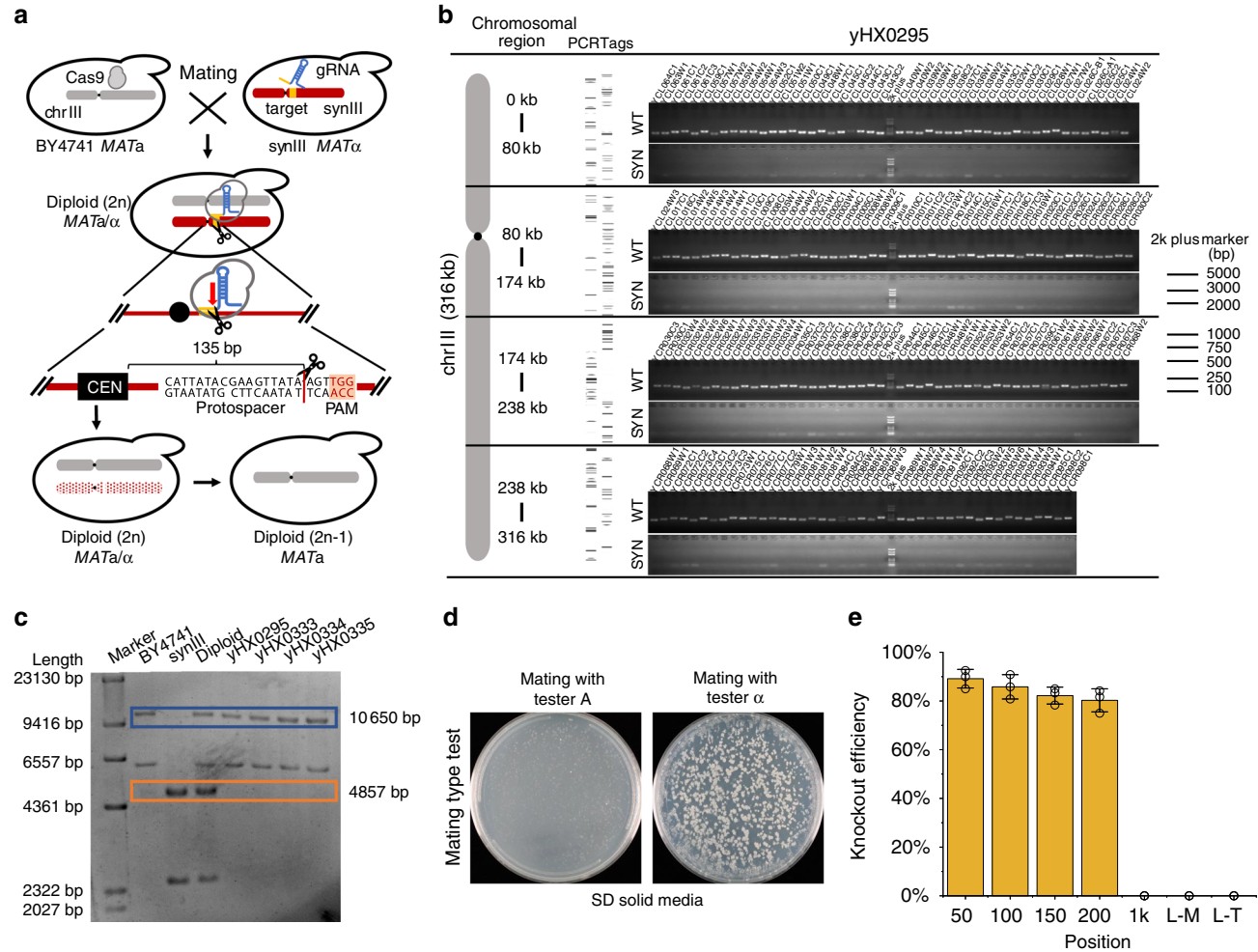

**Fig. 1 Chromosome elimination by CRISPR–Cas9 induced one DSB near the centromere. a** Mechanism for chromosome elimination via CRISPR–Cas9. The gray chromosome represents chrIII, and the red chromosome represents synIII. The CEN represents the centromere and the mosaic red chromosome represents the process of chromosome elimination in the diploid strain. **b** All synthetic and native PCRtags on synIII were tested. 372 synthetic and wild-type PCRtags were used to verify chromosome elimination of synIII in yHX0295. **c** Southern blot analysis of synIII elimination. The genomes of the BY4741, synIII, diploid (BY4741-synIII) were used as controls and four synIII elimination diploid strains (yHX0295, yHX0333, yHX0334, and yHX0334) were analyzed. All genomes were digested with HindIII. ChrIII yielded a 10,650 bp fragment, and synIII yielded a 4857 bp fragment after digestion and hybridization with the probe. The blue frame indicates fragments from chrIII, and the orange frame indicates fragments from synIII. **d** Mating type test of CRISPR-containing diploid strains by tester A and tester α. SD solid media with tester A or tester α were mated to media with CRISPR-containing diploid strains. **e** Efficiency of chromosome elimination at different positions of synX. L-M represents the middle of the left arm, and L-T represents the telomere of the left arm. A total of 120 strains were tested for each replicate and the error bars indicate the standard deviations of all ($n = 3$) biological replicates. Source data are provided as a Source Data file.

confirmed the elimination of synX, which encodes RFP, and the endoreduplication of chrX, which encodes GFP (Fig. 2a).

Furthermore, the synthetic violacein pathway was used as a colorimetric phenotype to visualize the process of chromosome drive. With respect to the BY4742 strain, a six-gene-member violacein pathway (*vio*A, *vio*B, *vio*C, *vio*D, *vio*E, and *URA3*) spanning a region of 13717 bp was inserted into the left arm of chrX, and the CRISPR–Cas9 expression cassette was inserted into the right arm of chrX (Supplementary Fig. 5). With respect to the synX strain, the synthetic CRISPR-targeted sequence (5′-GTTG CAAATGCTCCGTCGAC<u>GGG</u>-3′) was inserted 50 bp from the centromere on the left arm of synX, preventing potential transmission in wild-type yeast. The BY4742 and synX strains described above were subsequently mated to induce the process of chromosome drive (Fig. 2b). To investigate the driven genotype of the progeny, the CRISPR-containing diploid strains were subjected to whole-genome sequencing. The results revealed that

specific tag sequences of synX had been lost, and the sequencing depth of chrX was similar to that of the other chromosomes in the driven diploid strain yHX0296, verifying the elimination of the entire synX and the duplication of chrX (Fig. 2c, d).

To further investigate the driven phenotype of the progeny, CRISPR-containing diploid strains were randomly selected to undergo sporulation and tetrad dissection. Spores from the dissected tetrads were plated on synthetic complete (SC) media to visualize the color of colonies. A total of 70 individual diploid strains were randomly selected and tested. If the synX in the diploid strain was eliminated and the counterpart violacein-chrX was endoreduplicated, all four spores would display purple color, while the undriven diploids would undergo normal inheritance, producing two purple spores and two white spores per tetrad. The results showed that 87.14% (61/70) of the diploid strains inherited the violacein pathway, which was roughly equal to the efficiency of chromosome elimination for synX (Fig. 2e, Supplementary

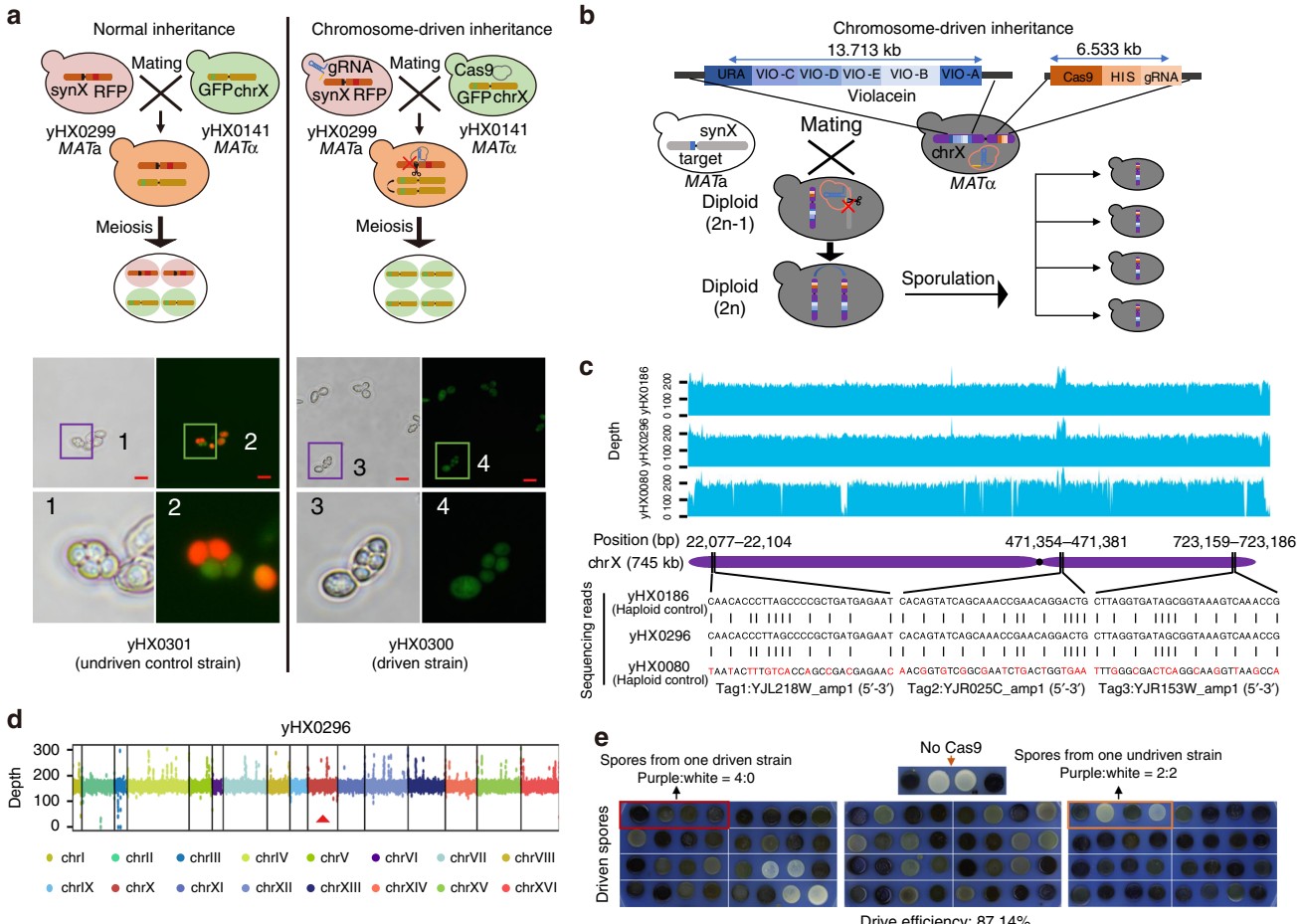

**Fig. 2 Chromosome drives with a synthetic violacein pathway. a** Verification of chromosome endoreduplication via GFP and RFP in yeast. Strains were identified under visible light and observed for GFP and RFP. The haploid controls yHX0299 and yHX0141 displayed red and green color, respectively, under a fluorescence microscope. Three tetrads from the one driven strain, yHX0300, displayed a green color in all four spores. One tetrad was displayed at a higher resolution. One tetrad from the diploid control yHX0301 displayed two green spores and two red spores. Numbered squares indicate single tetrad shown at a higher resolution in the below panels. Scale bar, 10 μm. **b** Workflow of the chromosome-driven inheritance with components of the synthetic violacein pathway. **c** Whole-genome sequencing analysis of synX elimination in yHX0296. All the PCRTags in yHX0296 matched with the haploid control (yHX0186), and three of them were displayed. The red bases represent the mismatched bases of PCRTags compared yHX0296 with the haploid control (yHX0080). **d** The coverage map of yHX0296. The driven diploid strain yHX0296 was used for the analysis of chromosome endoreduplication. The sequencing depths of all sixteen chromosomes were at the same scale. The sequencing depth of chrX was similar to that of the other chromosomes. The red triangle indicates the position of chrX in the coverage map. **e** Phenotypic test of spores from dissected tetrads. The spores with the synthetic violacein pathway produce purple colonies on SC media. In the absence of Cas9, normal 2:2 segregation was observed. A total of 87.14% of dissected tetrads were observed as having four purple spores. The spores of all 70 dissected tetrads are shown in Supplementary Fig. 6. The red frame indicates four spores from one driven strain and the orange frame indicates four spores from one undriven strain. Source data are provided as a Source Data file.

Fig. 6 and Supplementary Table 3). To further verify chromosome elimination, spores from one driven diploid strain were tested for the elimination of synX. PCRTag analysis of four spores derived from one tetrad showed that 12 of the synthetic PCRTags across synX were lost in spores yHX0157, yHX0158, yHX0159, and yHX0160, indicating the loss of the entire synX (Supplementary Fig. 7). Pulsed-field gel electrophoresis (PFGE) analysis also revealed that synX was lost and that chrX (encoding all the components of the violacein pathway) were present in all four spores (Supplementary Fig. 8). In all 61 driven tetrads tested, four identical spores with violacein pathways were obtained from each tetrad, further verifying the duplication of violacein-chrX in the driven tetrads. We also tested the stability of the chromosome drive through multiple rounds of mating and sporulation. As shown in Supplementary Fig. 9, the purple-driven strain can transmit violacein-chrX to white yeast cells within three rounds of experiments.

**Preferential transmission of complex genetic traits in yeast.** Our results from the above experiment demonstrate that chromosome drive enables biased inheritance on a chromosomal scale in *S. cerevisiae*. How can we utilize chromosome drive compared with other self-propagating drive systems? Using quantitative trait locus (QTL) mapping and genome-wide association studies (GWASs), researchers have shown that many traits are polygenically influenced[29]. Additionally, recent developments in synthetic biology have enabled the redesign and synthesis of biosynthetic pathways and whole chromosomes[25,26,30,31]. Thus, we attempted to demonstrate the ability of chromosome drive to transmit complex genetic traits via megabase synthetic chromosome XII (synXII), heterogenous Y12 chromosome XIV (chrXIV) and Synthetic Chromosome Rearrangement and Modification by LoxP-mediated Evolution (SCRaMbLE)d synX in *S. cerevisiae* in subsequent experiments.

With a highly repetitive ribosomal DNA (rDNA) gene cluster, chromosome XII (chrXII) is one of the largest chromosomes in the genome of *S. cerevisiae*. The internal transcribed spacer (ITS) is contained within the rDNA sequence and is considered the marker for species identification[32]. In a previous study, a megabase synXII of *S. cerevisiae* was chemically synthesized, and the rDNA was completely replaced by rDNA from *Saccharomyces bayanus* (*S. bayanus*) at the same position in JDY476[33]. Here, we used chromosome drive to transmit the *S. bayanus* rDNA on synXII to *S. cerevisiae* (Fig. 3a). In our study, the CRISPR-Cas9 expression cassette was inserted 242 kb from the centromere on the right arm of synXII in JDY476 cells (Supplementary Fig. 5). The synthetic CRISPR-targeted sequence (5′-GTTGCAAATGCTCCGTCGAC GGG-3′) was inserted 50 bp from the centromere of chrXII in BY4741. The process of chromosome drive was initiated by mating the two strains described above. PCRTag analysis of randomly selected diploid strains revealed that chrXII was eliminated in 20.83% (20/96) of the tested strains (Supplementary Table 3). Whole-genome sequencing analysis revealed that the wild-type PCRTags were lost and that the entire chrXII was eliminated in the diploid strains (yHX0297) (Supplementary Fig. 10). Owing to the difference in the ITS region between *S. bayanus* and *S. cerevisiae*, it can be used as a marker for synXII chromosome drive. Investigation of the driven spores via Sanger sequencing and restriction enzyme digestion of the PCR products of the ITS region indicated that the *Apa I* restriction site within the ITS region were no longer present in the tested driven strains (Fig. 3b, c). Thus, the rDNA in the driven strains had been swapped with the rDNA from *S. bayanus* by chromosome drive. These results indicated that chromosome drive can be used to transmit the specific chromosomal structures of repetitive rDNA gene cluster, resulting in changes in species identity.

Y12 is an industrial yeast strain used for fermentation owing to its thermotolerance[34]. However, it is difficult to dissect the explicit genetic mechanism underlying this thermotolerance since the strategy of adaptive evolution is commonly used to confer traits to industrial strains[35,36]. Here, using the strategy of chromosome elimination, we first attempted to map the thermotolerance trait at the chromosome scale in the heterozygous Y12-BY4741 diploid strain. Using a growth fitness test of haploid strains derived from the loss-of-heterozygosity (LOH) diploid strains at 40 °C, we found that chrXIV of Y12 contributes the most to thermotolerance (Supplementary Fig. 11). To further confirm whether the strain containing chrXIV of Y12 has a thermotolerance trait using chromosome transfer and elimination, we constructed a heterozygous haploid strain (yHX0266) in which chrXIV of BY4741 was replaced with chrXIV of Y12 (Supplementary Fig. 12). The results showed that yHX0266 was more thermotolerant than was BY4741, indicating that chrXIV of Y12 was the thermotolerance-related chromosome (Supplementary Fig. 13). To demonstrate the ability of chromosome drive to transmit the thermotolerance trait to other yeast strains, we attempted to transmit chrXIV of Y12 to laboratory yeast strain BY4742 via chromosome drive. As shown in Fig. 3d, yHX0266 was equipped with a driven CRISPR-Cas9 system, and the BY4742 strain was inserted with the synthetic CRISPR-targeted sequence (5′-GTTGCAAATGCTCCGTCGAC GGG-3′) 50 bp from the centromere of chrXIV (Supplementary Fig. 5). The process of chromosome drive was initiated by mating of the two strains as described above. Depending on the sequence differences between chrXIV of BY4742 and that of Y12, specific tags were designed to assess the elimination of the corresponding chromosome of BY4742 (Supplementary Fig. 14). PCR-based analysis revealed that of BY4742 chrXIV was

eliminated in 40.63% (39/96) of the tested diploid strains (Supplementary Table 3). To verify the complete elimination of BY4742 chrXIV and the transmission of Y12 chrXIV into the driven spores, four driven spores (yHX0285, yHX0286, yHX0287, and yHX0288) from one diploid strain were tested by whole-genome sequencing. On the basis of the distribution of single-nucleotide polymorphisms (SNPs) throughout the entire chrXIV, the pattern of four driven spores was the same as that of Y12, indicating successful transmission of the entire Y12 chrXIV to the spores (Fig. 3e and Supplementary Table 4). Further, we tested the growth fitness of the four driven spores at 30, 37, and 40 °C by serial dilution assays. The results showed that all four driven spores were obviously more thermotolerant than the BY4742 strain was, demonstrating the use of chromosome drive to transmit complex genetic traits in yeast (Fig. 3f).

As part of the Sc2.0 project, synthetic yeast chromosomes have been designed with hundreds of palindromic loxPsym sites located downstream of nonessential genes throughout the chromosomes[25,26]. Genomic segments between loxPsym sites can be rearranged with the expression of Cre recombinase, which leads to stochastic deletion, inversion, translocation, and duplication within and between synthetic chromosomes, which is referred to as the Synthetic Chromosome Rearrangement and Modification by LoxP-mediated Evolution (SCRaMbLE) system[37–40]. Under suitable screening conditions, SCRaMbLEd strains with improved traits can be generated[38,40]. However, owing to the process of totally random chromosomal rearrangement, it is still a challenge to dissect the structural variations that lead to specific phenotypic changes[38–40]. Here, we used chromosome drive to attempt to transmit the newly generated unresolved trait of chromium resistance. First, SCRaMbLEd synX strains were screened on SC media supplemented with 0.15 mM Cd (NO₃)₂. Through a serial dilution assay, one SCRaMbLEd strain, yHX0065, was selected on the basis of its visible improvement in chromium resistance, and it was used as a drive carrier to transmit SCRaMbLEd synX to laboratory yeast BY4742 (Supplementary Fig. 15). The yHX0065 strain was equipped with the driven CRISPR-Cas9 system, and the BY4742 strain was inserted with the synthetic CRISPR-targeted sequence (5′-GTTGCAAAT GCTCCGTCGAC GGG-3′) 50 bp from the centromere of chrX (Supplementary Fig. 5). The process of chromosome drive was initiated by mating the two strains as described above (Fig. 3g). PCRTag analysis revealed that the target chromosome chrX was eliminated in 87.5% (84/96) of the tested diploid strains (Supplementary Table. 3). Whole-genome sequencing analysis revealed that the chromosome-elimination strain yHX0332 contained only synthetic PCRTags, verifying the elimination of chrX in yHX0332 (Supplementary Fig. 16). One driven diploid was selected and sporulated. Synthetic PCRTag analysis of the dissected spores showed that all four spores had been transmitted together with synX (Supplementary Fig. 17). Electrophoretic karyotype analysis of the dissected spores by PFGE revealed that all four spores were driven together with SCRaMbLEd synX, which was the same as that of strain yHX0065. It was also shown that SCRaMbLEd synX was longer than the original synX, indicating structural variation in synX induced by SCRaMbLE (Fig. 3h). To test the capability of chromium resistance of the driven spores, the four spores were subjected to a serial dilution assay on SC media supplemented with 0.15 mM Cd(NO₃)₂. The results showed that all driven spores grew better than the BY4742 strain did on the chromium media, indicating successful transmission of the unresolved phenotype of chromium resistance by chromosome drive (Fig. 3i). It was also indicated that the SCRaMbLEd synX of strain yHX0065 is directly related to chromium resistance.

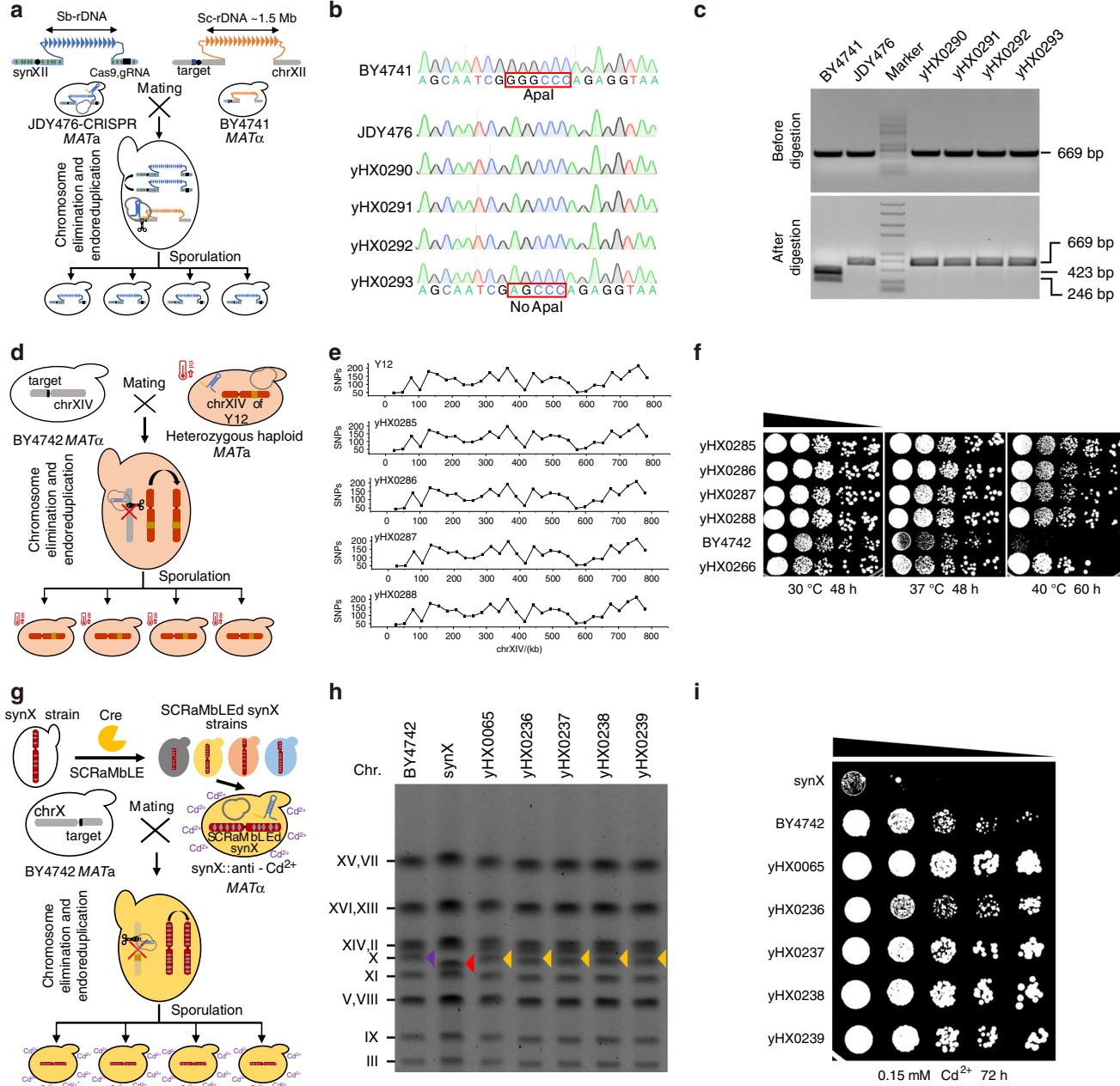

**Fig. 3 Preferential transmission of complex genetic traits in yeast via chromosome drive. a** Process by which chromosome drive transmits synXII by mating and sporulation. SynXII is encoded by a, highly repetitive rDNA sequence of *S. bayanus*. **b** Sequences of PCR products amplified from ITS regions for BY4741, JDY476 and four driven spores. The red frame indicates the sequence difference in rDNA between *S. cerevisiae* and *S. bayanus*. **c** Restriction enzyme digestion to verify the absence of the native ITS sequence in the driven spores. The PCR products were treated with (bottom) or without (top) *Apa*I. Four driven spores (yHX0290, yHX0291, yHX0292, and yHX0293) was used for the test. BY4741 and JDY476 were used as negative and positive controls, respectively. **d** Process by which chromosome drive transmits thermotolerance-related chromosomes. The detailed process of the construction of the heterozygous haploid is shown in Supplementary Fig. 12. **e** Distribution of SNPs throughout the entire chrXIV for four driven spores (yHX0285, yHX0286, yHX0287, and yHX0288) and Y12. The number of SNPs was counted every 26 kb, with chrXIV of BY4741 used as a control. **f** Serial dilution assay of four driven spores (yHX0285, yHX0286, yHX0287, and yHX0288) at 30, 37, and 40 °C. BY4742 and yHX0266 were used as controls. **g** Process by which chromosome drive transmits SCRaMbLEd synX with an unresolved chromium resistance trait. **h** Electrophoretic karyotype analysis of four dissected spores (yHX0236, yHX0237, yHX238, and yHX0239) via PFGE. Strains BY4742, synX, and yHX0065 were used as controls for chrX, synX, and SCRaMbLEd synX, respectively. The purple triangle indicates the positions of chrX, the red triangle indicates the positions of synX and the yellow triangle indicates the positions of SCRaMbLEd synX. **i** Serial dilution assay of four driven spores (yHX0236, yHX0237, yHX238, and yHX0239) under 0.15 mM Cd (NO₃)₂ for 72 h. Strains synX and BY4742 were used as controls. Source data are provided as a Source Data file.

## Discussion

The chromosome drive system is designed to transmit a desired chromosome via sexual reproduction. Chromosome elimination is a vital step in the CRISPR-Cas9 chromosome drive system. In this study, we found that only one DSB near the centromere induced by CRISPR-Cas9 can effectively eliminate a chromosome in *S. cerevisiae*. In this species, DSB repair is based on mainly homology-directed repair (HDR) rather than nonhomologous

end-joining (NHEJ) in the presence of a homologous repair template[41]. It was reported that the efficiency of HDR around centromeres is lower than that in other regions of chromosomes in *S. cerevisiae*[42–45]. Thus, we speculated that the CRISPR-Cas9-induced DSB near the centromere may be hard to repair by HDR or NHEJ. The broken chromosome would be unstable and ultimately lost in the yeast cells. In other organisms, the contribution of HDR and NHEJ to DSB repair is quite different. In mammals, NHEJ is considered the major way to repair DSBs. For species with regional centromeres, previous studies have revealed that an entire chromosome can be eliminated by CRISPR-Cas9 targeting at multiple specific sites on the chromosome or in the centromeric region[46–48]. In addition, we also noticed that the elimination efficiency for different chromosomes varied in *S. cerevisiae* (Supplementary Table 3). The off-target mutations were then characterized via whole-genome sequencing. After analyzing ten individual off-target strains for each drive experiment of synX, chrXII, chrXIV, and chrX, we found that HDR happened in the CRISPR target region for the majority of the off-target strains in the experiments involving synX, chrXIV, and chrX. However, the CRISPR-targeted sequence remained intact for nine of the ten off-target strains in the drive experiment of chrXII (Supplementary Table 5). These results may be due to the low efficiency of CRISPR for chrXII, which contains a highly repetitive ribosomal gene cluster for the formation of a nucleolar structure, and there is also evidence that the efficiency of DSBs varies within different chromosomes[45]. Regarding of the existence off-target strains, it may be a problem for multiple generations of chromosome elimination. The chromosome drive system enables biased inheritance of an entire desired chromosome by elimination of the target chromosome and subsequent autonomous chromosome endoreduplication. In *S. cerevisiae*, chromosome endoreduplication is a recovery mechanism for the chromosome elimination[28,49]. Evidence has also shown that uniparental disomy (UPD), the phenomenon in which two copies of a chromosome from one parent are present and no copy from the other parent is present, occurs via monosomy rescue of chromosome XIV and chromosome XV[50,51]. However, in some cases, chromosome endoreduplication may not happen, and the mechanism underlying the chromosome endoreduplication essentially remains unknown[28]. To achieve the biased inheritance of desired chromosomes via the chromosome drive system, the process of chromosome endoreduplication may not be critical since elimination of the target chromosome in the zygotes results in inviable progeny without the desired chromosome. This process employs the same principle as that of CRISPR toxin-antidote drive, in which recessive lethal alleles are formed and half of offspring with recoded synthetic genes are rescued[18,19,52].

Chromosome elimination via CRISPR–Cas9 is a powerful tool to induce whole-chromosomal LOH and can be developed as a method for mapping specific chromosome-related traits. In addition, chromosome elimination via CRISPR-Cas9 also provides a strategy for consolidating multiple synthetic yeast chromosomes into one cell by knocking out the counterpart wild-type chromosomes in synthetic-wild type heterozygous diploid strains, ultimately leading to the construction of a synthetic yeast genome[25,26,30,33,49]. With sexual reproduction and desired chromosome endoreduplication, the chromosome drive system can provide a method for the transmission of chromosome-related traits and various unresolved traits to facilitate genetic engineering of industrial strains. In this study, we demonstrate that the chromosome drive system enables biased inheritance of complex genetic traits or unresolved genetic traits on a chromosomal scale in yeast, extending possible applications of self-propagating drive. With respect to the chromosome drive system in other given species, many studies on the efficiency of

chromosome elimination, fertility and biosafety consequences should be performed.

## Methods

**Strains, plasmids, and primers**. All the yeast strains used in this study are listed in Supplementary Data 1 the plasmids used are listed in Supplementary Data 2, and primers used are listed in Supplementary Data 3.

**Yeast transformation**. The yeast colonies were incubated in 5 mL of yeast peptone dextrose (YPD) overnight at 30 °C with shaking at 220 rpm. Afterward, 200 μL of the culture was transferred to 5 mL of YPD. When the OD$_{600}$ reached 0.4–0.6, 1 mL of the cells was harvested and washed twice with sterile H$_2$O. The pellets were subsequently treated with 1 mL of 0.1 M LiOAc solution and then gently placed on ice for 5–10 min. Afterward, 620 μL of polyethylene glycol (PEG) 3350, 40 μL of salmon DNA and 90 μL of 1 M LiOAc were mixed together, which served as transformation buffer. The salmon DNA was preheated at 100 °C for 12 min and then placed on ice to cool. Pretreated cells (150 μL) were immersed in transformation buffer and then mixed by being inverted up and down 7–10 times, after which they were cultured at 30 °C for 30 min. The culture was then mixed with dimethyl sulfoxide (DMSO) and heat-shocked at 42 °C for 18 min. The solution was centrifuged at 1500 × *g* for 2 min, after which the supernatant was decanted. The cells were recovered and placed in 400 μL of 5 mM CaCl$_2$ for 7–10 min at room temperature, after which they were plated onto corresponding selective media.

**Mating with tester A and tester α**. Two different mating-type strains lacking *HIS1*, referred to as tester A and tester α, were used. Both strains were selected and cultured overnight in 5 mL of YPD overnight at 30 °C with shaking 220 rpm. Two hundred microliter of both culture solutions was plated onto SD solid media and recorded as test plates. Moreover, the diploid strains that remained to be tested for mating type were copied onto test plates, respectively. The SD solid media were cultured at 30 °C for 24 h to observe the growth of the colonies. The mating type of Diploid strains that can grow on test plates was A, that are unable to grow on the test plates was A/α.

**Yeast mating process**. Yeasts with different mating types were incubated in corresponding selective media overnight. 1 mL of yeasts with different mating types was harvested and washed twice with sterile ddH$_2$O twice. Two hundred microliter of each culture was then added to 5 mL of YPD (in the YPD, each strain grew well) and incubated at 30 °C with shaking at 220 rpm for 8 h. One milliliter of the mating solution was subsequently harvested and washed with sterile ddH$_2$O twice. Ten microliters of solution and 200 μL of sterile ddH$_2$O were mixed together, plated onto selective media, and then incubated at 30 °C for 2–3 days.

**PCRTag analysis of chromosome elimination**. PCRTags located at both ends of the chromosome and at both sides of the centromere on each chromosome were selected as markers to assess the efficiency of syn X, synIII, chrX, synXII, and chrXIV elimination. The loss of PCRTags of synX, chrX (YJL219W_amp1, YJL002C_amp1, YJR001W_amp2, YJR155W_amp1), synIII (YCL064C_amp1, YCL001W_amp1, YCR002C_amp1, YCR098C_amp1), and synXII (YLL048-C_amp1, YLL001W_amp1, YLR001C_amp1, YLR455W_amp1) can be used to assess the elimination of chromosomes. With respect to the efficiency of chrXIV elimination, PCRTags (Y12-left, Y12-right) were designed to test chromosome elimination.

**Meiosis and sporulation**. Yeasts were incubated in 5 mL of YPD overnight at 30 °C with shaking at 220 rpm, after which 200 μL of the culture was transferred to 5 mL of 2× YPD and then incubated at 30 °C with shaking at 220 rpm for 10 h. All the cells were harvested and washed with sterile ddH$_2$O three times (the YPD was washed away totally to ensure the efficiency of sporulation). One milliliter of 50× sporulation media, 500 μL of the required amino acids (100×; uracil, histidine, leucine, or tryptophan), and 150 μL of 10% yeast extract were mixed together and then diluted with sterile ddH$_2$O to a volume of 50 mL. All washed cells were transferred into sporulation solution and mixed well. The culture was incubated at 25 °C with shaking at 200 rpm for 3–6 days. The presence of tetrads was subsequently observed with a microscope.

**Serial dilution assays for selections**. Yeasts were incubated in 5 mL of YPD overnight at 30 °C with shaking at 220 rpm, after which 200 μL of the culture was transferred to 5 mL of YPD at 30 °C with shaking at 220 rpm and then incubated to an OD$_{600}$ of 1. The solution was serially diluted in 10-fold increments in ddH$_2$O five times, and 4 μL of the diluted solution was plated from lowest to highest concentration on the corresponding media. When the solution was dry, the media were maintained at a suitable temperature for 2–3 days.

**Pulsed-field gel electrophoresis (PFGE)**. The PFGE protocol was modified[53], and a single colony was inoculated into 5 mL of YPD overnight, with shaking at

30 °C. One milliliter of the overnight culture was transferred to a tube and was centrifuged at $1200 \times g$ for 2 min at room temperature. Cells were washed twice in solution I (0.05 M EDTA, 0.01 M Tris, pH 7.5) and resuspend in 150 µL solution I with 10 µL zymolase (2 mg mL$^{-1}$ zymolase 20 T, 10 mM Sodium Phosphate, pH 7.5). Cells were placed in the 42 °C heat block. Two hundred and fifty microliter of agarose solution (1% (w v$^{-1}$) low melting temp agarose, 0.125 M EDTA, pH 7.5) was pre-incubated at 42 °C and mixed with cells by pipetting with a wide bore pipette tip in the tube. The tube was placed on ice immediately and added 400 µL of LET (0.5 M EDTA, 0.01 M Tris, pH 7.5). The tube was incubated for 8–10 h overnight at 37 °C, and placed on ice for 10–20 min, then transferred the agarose plug in the tube to a 15 ml falcon tube. Four hundred microliter of NDS (0.5 M EDTA, 0.01 M Tris, 1% (w v$^{-1}$) Sodium Lauryl Sarcosine, 2 mg mL$^{-1}$ proteinase K, pH 7.5) was added and incubated overnight at 50 °C. The tubes was placed on ice for 10 min. The NDS was exchanged for solution I and rock/swirl gently at room temperature for 1 h and repeated the wash three more times. Plugs were stored in fresh solution I at 4 °C. The electrophoresis process was performed in one stage, and the gel was made with 1% low-melting agarose and 1× Tris/borate/EDTA (TBE) buffer. The electrophoresis conditions were as follows: switch time of 60–120 s, run time of 20 h, angle of 120° and voltage of 6 V cm$^{-1}$.

**Fluorescence detection of tetrads.** The strains were sporulated prior to observations. Ten microliters of sporulated culture was mounted onto a slide, which was then sealed with a coverslip. The fluorescence of the tetrads was scored via a Nikon Eclipse Ci-L Scope equipped with Nikon Plan Fluor 40×/0.75 objective with Nikon DAPI-FITC filter and Nikon TRITC filters. The tetrads were observed under visible light and then scored for RFP and GFP. The images were taken under suitable light conditions.

**Thermotolerance-related chromosomes mapping.** The synthetic CRISPR-targeted sequence (5′-GTTGCAAATGCTCCGTCGACGGG-3′) was inserted 50 bp from the centromere of each chromosome of BY4741. In addition, *URA3* was integrated into chrIII in each strain. In the Y12 strain (*MATα*), components of the CRISPR-Cas9 system were inserted into chrXIV. The two strains described above were mated and plated on solid SC-URA-LEU media, after which they were cultured at 30 °C for 3–5 days. Single colonies were selected to test chromosome elimination in the heterozygous Y12-BY4741 diploid strain via PCR-based analysis. The PCRTags used are described in Supplementary Table. 8. The chromosome-eliminated diploid strains were selected for sporulation and dissection into spores. The spores were tested for their thermotolerance with Y12 and BY4741 via a serial dilution assay at 30 and 40 °C. The spores that contained thermotolerance-related chromosomes of Y12 grew better than the controls did at 40 °C.

**Construction of a thermotolerant heterozygous haploid.** The *karI* gene is an essential gene in *S. cerevisiae*. In yeast, the *karI* mutation causes a lack of nuclear fusion. When one of the mating partners has a *karI* mutation, nuclear fusion is unable to occur[54]. First, we constructed a *karI* mutation strain to transfer chromosomes. We deleted *karI* followed by the complement of the plasmid containing *karI* in BY4741. For the plasmid containing *karI*, *prs415* was extracted from *Escherichia coli*, and the *karI* DNA segment was amplified from the yeast genome. Both of them had *NotI* and *BamHI* restriction sites and were digested. The *karI* DNA segment was fused to the *prs415* carrier using T4 DNA ligase to construct the plasmid. The *karI*-deleted segment was constructed together with upstream and downstream homologous arms and the *URA3* marker. The deleted segment and plasmid were cotransformed into BY4741, which was then plated onto solid SC-URA-LEU media. The media were subsequently incubated at 30 °C for 2–3 days. The transformed strains were verified by PCR, and the correct strain was selected for subsequent construction. The *karIΔ15* segment was transformed into the positive strain, plated on solid SC + 5-FOA media where strains lacking the URA3 marker can grow, and then incubated at 30 °C for 2–4 days. Colonies that grew on solid SC + 5-FOA were analyzed via PCR, and the PCR products of the colonies were aligned with the sequence of *karIΔ15* to obtain the *karIΔ15* mutation strain (yHX0230).

In yHX0230, the CRISPR-targeted sequence (5′-GTTGCAAATGCTCCGT CGACGGG-3′) was inserted 50 bp from the centromere on the left arm of chrXIV by the use of the *URA3* marker, and the *LEU2* marker was integrated into chromosome XI. In addition, *can-100* was deleted in yHX0230. In the Y12 strain yHX0192 (*MATα*), the CRISPR-Cas9 system was inserted into chrXIV. Afterward, yHX0230 and yHX0192 were mated, and all the mated cells were plated onto solid SC-URA-LEU+ canavanine media. The media were then cultured at 30 °C for 3–5 days. Single colonies were selected to test the mating type and the loss of chrXIV in BY4741 via PCR-based analysis. In heterozygous haploids, the mating type was *MATa*, and chrXIV of BY4741 was eliminated.

**Whole-genome sequencing.** The strain samples were prepared for whole-genome sequencing according to GENEWIZ's standard preparation protocol. The samples were sent on dry ice to GENEWIZ in Suzhou. The sequencing data were analyzed and inductively charted for easy analysis.

**SCRaMbLE of synX strain to generate chromium-resistance.** The Cre-EBD plasmid, in which an engineered Cre recombinase was fused to the murine estrogen-binding domain, was used to prevent Cre expressed in the absence of β-estradiol and the activity of SCRaMbLE. The synX strains carrying the Cre-EBD plasmid were incubated in 5 mL of SC-HIS media overnight at 30 °C and 220 rpm. Five microliter of β-estradiol (1 mM) and the culture was were then mixed together and cultured at 30 °C with shaking at 220 rpm for 8 h. One microliter of the cells was harvested and washed with ddH$_2$O twice to remove all the estradiol. The solution was subsequently diluted 1000-fold, and 100 µL of the diluted solution was plated onto solid SC media supplemented with 0.15 mM Cd(NO$_3$)$_2$. The media were then cultured at 30 °C for 3–5 days, after which the colony sizes were measured.

**Reporting summary.** Further information on research design is available in the Nature Research Reporting Summary linked to this article.

## Data availability
All sequencing data that support the findings of this study have been deposited into GeneBank that are available from the Sequence Read Archive under Accession Code SRP269526. Source data are provided with this paper. Any other relevant data are available from the authors upon reasonable request.

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

## Acknowledgements

We are grateful to Junbiao Dai for providing synXII strain, Jef D. Boeke and Srinivasan Chandrasegaran for providing synIII strain. This work was supported by the National Key Research and Development Program of China (2018YFA0900100), the National Natural Science Foundation of China (31971351 and 21750001) and Young Elite Scientist Sponsorship Program by CAST (YESS) (2018QNRC001).

## Author contributions

Y.W. and Y.J.Y. conceived the overall project; H.X. and Y.W. designed the experiments; H.X., M.Z.H., and S.Y.Z. performed the experiments; H.X. and M.Z.H. performed bioinformatic analysis; H.X., B.Z.L., Y.J.Y., and Y.W. wrote the paper, with input from all authors.

## Competing interests

The authors declare no competing interests.
