## [Peer Review File · Nature Communications]

Reviewers' Comments:

Reviewer #1:

Remarks to the Author:

The authors outline a method to introduce episomal CRISPR components into yeast to enable biased inheritance of specific chromosomes. These systems include an episomal plasmid system capable of cleaving near the centromere, thereby preventing the inheritance of this chromosome in the progeny, and they demonstrate this is dependent on the proximity of the cleavage site with the centromere (Figure 1). They further use this system to demonstrate biased inheritance of a chromosome encoding genes required for the violetin pathway as a reporter. While they postulate this is due to a duplication of the Violetin-containing chromosome, we do not agree that this conclusively represents a duplication or if it is due to simple destruction and absence of the non-engineered chromosome (Figure 2). In Figure 3 they outlined the modification of the system to enable biased inheritance of a set of species-specific ribosomal loci, and in Figure 4 they outline biased inheritance of a thermotolerant trait. In Figure 5 they use the system to elucidate location of a previously-uncharacterized trait of chromium resistance in an engineered SCRaMbLED yeast system.

Major corrections:

- We understand this is likely an ESL manuscript. However throughout the manuscript there are significant grammatical mistakes, and improper use of scientific language that greatly hinders reader comprehension. We therefore recommend seeking proofing from a 3rd party, following such edits the intention of the authors will be much more clear.
- Because the proposed system relies on destruction of the wild type chromosome (not the duplication or reproduction of the synthetic genetic element) should it be considered a true gene drive? Perhaps use of the term Meiotic drive would be more appropriate for this system, as it relies on destruction of the wt chromosome. However, in a traditional sense, drives are generally reserved for those expressing CRISPR components chromosomally, not necessarily episomally. Can the authors justify this design decision?
- We would like to see a transgene map for the Cas9 and gRNAs throughout the work.
- Line 30: Reference #4 is not necessarily what we would consider the primary, and most notable, literature outlining the biology of Selfish genetic elements. Can the authors justify their reasoning for selecting this specific reference for inclusion here? Or identify more relevant references to cite in its stead
- The authors speculate that the entire chromosome is being driven (Figure 1). However, can recombination be ruled out? Are there markers on the engineered chromosome – distal to the CRISPR site – that could be assayed for to determine if the entire chromosome 'drove' as a single unit?
- Figure 1: It would be nice to see an overview map of the CRISPR cassette itself.
- Figure 1A. It would be nice if it were more clear that the orange chromosome was being destroyed in the figure. Currently, with the alignment between the grey and orange chromosomes, it appears as if they are oriented for homologous recombination – not destruction.
- Figure 1B: A number of comments: The upper PCR gels in each panel (syn-PCRTag) lack a positive control. Most gels lack the required positive or negative controls throughout the work unfortunately. All gels should either include a ladder or a label of fragment size. The bands of these gels are sufficiently faint it is difficult to be sure they are truly PCR-positive as the authors suggest. And can the authors address why the band sizes are all different in these PCR fragments?
- The authors should discuss the choice of Cas9 expression plasmid and its properties, if not in the results section, it would be nice to be given information on its properties in the methods section. The Methods section refer to Table 5 (which should also read "Supplementary Table 5" if it is to remain in the supplement), however Table 5 does not give detailed information on the plasmid properties.
- Line 67: While the SynIII marker appears to be absent by PCR (assuming a positive control would have worked), this fact alone cannot prove that the entire chromosome has been ablated. If the authors were to PCR for other markers distal to the SynIII site, then this may be able to lend

credence to their claims, however it would still not be sufficient for complete proof of the chromosome's absence. The claims outlined here require further support. While it appears this is addressed in Figure 2, the aforementioned claims can not be supported by the findings in Figure 1 alone.

- Line 71-74: the genetics described is unclear here.
- Line 110: How sure can we be that it has duplicated? Or is there just only one viable chromosome available, and the other mostly cleaved? Did they count the total number of tetrads that came out of the experiment?
- With the CRISPR components expressed episomally, could this be considered a drive? If they were to insert the CRISPR components into the genome, it could be considered a selfish chromosome.
- It would be a welcome addition for the authors to further elaborate on the practical application of such a system in the Discussion. Perhaps in industrial or agricultural applications, beyond mapping traits in a laboratory setting.
- Line 230: I don't know if they can make an association between their system and homology-directed repair. I don't believe they sufficiently demonstrated that homing was occurring, nor that the chromosome was duplicating (however I do believe their system is successfully preventing the cleaved chromosome from being inherited – however this is not the same as duplication).
-

Minor Corrections:

- Line 11: add "The" in front of target chromosome and "the" before desired chromosome for proper use of English grammar.
- Line 12: inclusion of the word "Whole" is considered inappropriate scientific language.
- We understand this is likely an ESL manuscript. However there are significant grammatical mistakes, and improper use of scientific language throughout hindering reader comprehension. This review will therefore not seek to correct for grammar, instead focusing the remainder of the review on Scientific merit and larger-scale errors under the assumption it will be proofed significantly prior to publication.
- The sentence beginning in line 18 is a run-on. Please shorten for clarity.
- Figure 1A: denote exact distance from centromere for precision (ie, not <200bp)
- Line 28: specify that it is "normal Mendelian inheritance" for accuracy and specificity.
- Line 33: reference 5 is incorrect
- Line 34: References 6-9 are exclusively mosquito gene drive papers, but the authors mention yeast and mammals as well. They should add or modify the references cited to reflect this.
- Line 40: It is not clear what the intended message of the sentence beginning with "It is still a challenge..." is. Please rephrase for clarity.
- Line 63: the gRNA target sequence should include the 5' and 3' annotations. And the PAM should be distinguished in some way. This should be done for all mentions of gRNA target sequence and primers throughout the text.
- Line 72: figure should be capitalized.

Reviewer #2:

Remarks to the Author:

In this study, the authors design a new type of gene drive system in yeast called a chromosomal drive. Cas9 and gRNA cut a site near the centromere of a chromosome, resulting in destruction of the chromosome. The drive chromosome, however, does not contain the target site, thus biasing inheritance toward the drive.

The toxin-antidote drive strategy was quite interesting. The paper will probably be impactful to the field and a more general audience. However, I do have some concerns that I believe must be addressed before the manuscript can be published. To give some perspective for this review, I

specialize in gene drive but not in yeast genetics.

Major Points

1. The entire manuscript requires a thorough edit for proper language. This will be necessary to publish the manuscript in an English-language journal. To help, here is a version of the abstract that has been minimally revised to correct several English errors:

“Self-propagating drive systems are capable of causing non-Mendelian inheritance and altering traits of populations. Here, we report a drive system called chromosome drive that eliminates a target chromosome and spreads a desired chromosome via CRISPR/Cas9. In our study, we found that a whole *Saccharomyces cerevisiae* chromosome can be eliminated efficiently through only one double strand break around the centromere using CRISPR/Cas9. As a proof-of-principle experiment of the CRISPR/Cas9 chromosome drives system, the synthetic yeast chromosome X was completely eliminated, and the counterpart wild-type chromosome X harboring a synthetic violetin pathway was driven throughout a population by sexual reproduction. We also demonstrate the use of chromosome drives to spread complex genetic traits in yeast. Megabase synthetic chromosome XII, heterogenous Y12 chromosome XIV, and SCRaMbLEd chromosome X were had biased transmission to progeny by chromosome drives in a population of *Saccharomyces cerevisiae* to spread a species identified ribosomal gene cluster from *Saccharomyces bayanus*, complex traits of thermotolerance, and unresolved traits of chromium resistance, respectively. Chromosome drives enables biased inheritance on a chromosomal scale, facilitating chromosome scale genetic mapping and extending applications of self-propagating drive.”

2. Why does Figure 1B and 1C seem to imply 100% chromosome elimination, but then the experiments in Figure 3C indicate that elimination is usually 80-90% when the site is sufficiently close to the centromere?

3. Why weren't the yeast able to do end-joining repair for the Cas9-induced breaks. Is end-joining somehow inhibited near centromeres? If so, is this also true in other species? These matters should be included in the manuscript.

4. The experimental details in Figure 2D are unclear, and I am unclear how to interpret the results of the spot picture. In general, I understand that the chromosome drive increased in frequency over three rounds or generations of mating, but I'm unclear what the starting and ending frequencies of the drive were, which is quite important for this sort of experiment.

5. Can the authors comment why the deletion efficiency is so different between the different strains in Supplemental Table 2? Is the cut rate lower, or did DNA repair (HDR or end-joining) occur? This might require a quick sequencing experiment, but this is critical for understanding what is happening in these yeasts.

6. Is there no recombination between chromosomes during yeast sexual reproduction? This could have an important effect on a chromosome drive and should be mentioned in the article if true (and discussed if not) to make it more accessible for non-specialist readers.

Minor Points

1. References 6 and 13 are the same. In general, in the introduction, it might also be useful to include several other CRISPR/Cas9 gene drives studies covering yeast and *Drosophila* (including a safeguarding study that used synthetic target sites).

2. In the first results subsection, it would be useful to make it explicit that the WT chromosome does not have the gRNA target site.

3. Figure 1B is redundant. It can probably be eliminated, since all the information is in Supplementary Figure 1. However, why was the band for the wt-PCRTag not present in a few samples? Also, "Left-M" and "T" should be defined in the legend for Figure 1D.

4. Supplemental Table 1 would be better if it had percentages, while still specifying the sample size of 120.

5. The idea of targeting an important site with CRISPR with a desired site is immune to cleavage has been explored as a gene drive in *Drosophila*. These drives themselves are quite different, but they share the same "toxin-antidote" principle. Indeed, the authors in the discussion say: "To achieve the purpose of biased inheritance of desired chromosomes by chromosome drive system, the process of chromosome endoreduplication may not be critical since elimination of target chromosome in the zygotes will result in inviability of progenies without desired chromosome." This is the main method of previous CRISPR toxin-antidote drives and proposals. It might be worth including them in the introduction and/or discussion:
<https://www.nature.com/articles/s41467-020-14960-3>
<https://bmcbiol.biomedcentral.com/articles/10.1186/s12915-020-0761-2>
<https://www.biorxiv.org/content/10.1101/861435v2>

Reviewers' comments:

Reviewer #1(Remarks to the Author):

Summary:

The authors outline a method to introduce episomal CRISPR components into yeast to enable biased inheritance of specific chromosomes. These systems include an episomal plasmid system capable of cleaving near the centromere, thereby preventing the inheritance of this chromosome in the progeny, and they demonstrate this is dependent on the proximity of the cleavage site with the centromere (Figure 1). They further use this system to demonstrate biased inheritance of a chromosome encoding genes required for the violetin pathway as a reporter. While they postulate this is due to a duplication of the Violetin-containing chromosome, we do not agree that this conclusively represents a duplication or if it is due to simple destruction and absence of the non-engineered chromosome (Figure 2). In Figure 3 they outlined the modification of the system to enable biased inheritance of a set of species-specific ribosomal loci, and in Figure 4 they outline biased inheritance of a thermotolerant trait. In Figure 5 they use the system to elucidate location of a previously-uncharacterized trait of chromium resistance in an engineered SCRaMbLED yeast system.

Major Concerns:

1. We understand this is likely an ESL manuscript. However throughout the manuscript there are significant grammatical mistakes, and improper use of scientific language that greatly hinders reader comprehension. We therefore recommend seeking proofing from a 3rd party, following such edits the intention of the authors will be much more clear.

Response: The entire manuscript was edited for grammar, phrasing, and punctuation to improve the flow and readability by Springer Nature Author Services.

2. Because the proposed system relies on destruction of the wild type chromosome (not the duplication or reproduction of the synthetic genetic element) should it be considered a true gene drive? Perhaps use of the term Meiotic drive would be more appropriate for this system, as it relies on destruction of the wt chromosome. However, in a traditional sense, drives are generally reserved for those expressing CRISPR components chromosomally, not necessarily episomally. Can the authors justify this design decision?

Response: We realized that meiotic drive and the chromosome drive have certain overlaps, leading to target chromosome inheritance with high efficiency compared with that of normal Mendelian inheritance. Thus, we classified chromosome drive as a meiotic drive and compared it with gene drive in the discussion. However, we indeed observed duplications of the homologous chromosomes after chromosome elimination in the diploid strains. A straightforward evidence of chromosome endoreduplication was illustrated by a fluorescence experiment through the process of chromosome drives (Figure 2A). Also, sequencing depth of duplicated chromosomes were used as the evidence for chromosome endoreduplication (Figure 2D). In initial experiment of synIII elimination, we expressed Cas9 and gRNA episomally for biosecurity considerations (Figure 1). Then, in drive experiments of synX, chrXII, chrXIV and chrX, CRISPR components were integrated into the corresponding chromosome (Figure 2; Figure 3). The specific transgene maps are shown in Supplementary Figure 5.

3. We would like to see a transgene map for the Cas9 and gRNAs throughout the work.

Response: We added transgene maps for all the Cas9 and gRNAs throughout the work in Supplementary Figure 5.

4. Line 30: Reference #4 is not necessarily what we would consider the primary, and most notable, literature outlining the biology of Selfish genetic elements. Can the authors justify their reasoning for selecting this specific reference for inclusion here? Or identify more relevant references to cite in its stead

Response: Thank you for your suggestion, we replaced Reference #4 with four more relevant references.

5. The authors speculate that the entire chromosome is being driven (Figure 1). However, can recombination be ruled out? Are there markers on the engineered chromosome – distal to the CRISPR site – that could be assayed for to determine if the entire chromosome 'drove' as a single unit?

Response: There is no recombination between the wild-type chromosome chrIII and synthetic chromosome synIII, because the elimination of synIII occurred before the process of meiosis, leading to loss of heterozygosity (LOH) in the diploid strains (Figure 1B; Figure 1C; Figure 1D; Supplementary Figure 2; Supplementary Figure 3).

There are indeed markers distal to the CRISPR site on the synthetic chromosome. Numerous gene-associated synthetic PCRTags were inserted throughout the synthetic chromosome, thus, could be assayed to determine whether an entire chromosome was driven as a single unit (Figure 1B; Supplementary Figure 2). In fluorescence experiment, the red fluorescence protein (156kb from the CRISPR site) was used as the marker to characterize the elimination of synX (Figure 2A). In addition, Southern blot and whole-genome sequencing were used to analyze the sequence of the driven strains. The results showed that the entire synIII was eliminated (Figure 1C; Supplementary Figure 3; Supplementary 4).

6. Figure 1: It would be nice to see an overview map of the CRISPR cassette itself.

Response: Details of all the CRISPR cassettes were added in Supplementary Table 7.

7. Figure 1A. It would be nice if it were more clear that the orange chromosome was being destroyed in the figure. Currently, with the alignment between the grey and orange chromosomes, it appears as if they are oriented for homologous recombination – not destruction.

Response: Thank you for your suggestions. We changed color of the chromosomes and highlighted the CRISPR targets to demonstrate the entire chromosome was destroyed in Figure 1A.

8. Figure 1B: A number of comments: The upper PCR gels in each panel (syn-PCRTag) lack a positive control. Most gels lack the required positive or negative controls throughout the work unfortunately. All gels should either include a ladder or a label of fragment size. The bands of these gels are sufficiently faint it is difficult to be sure they are truly PCR-positive as the authors suggest. And can the authors address why the band sizes are all different in these PCR fragments?

Response: We replaced the PCR gels with addition of a negative control, a positive control and a ladder in figure 1B and supplementary figure 2. The length of PCR products varies between 200 bp and 500 bp based on the locations of specific PCRTags sequences. All lengths and sequences of the PCRTags are listed in Supplementary Table 9.

9. The authors should discuss the choice of Cas9 expression plasmid and its properties, if not in the results section, it would be nice to be given information on its properties in the methods section. The Methods section refer to Table 5 (which should also read “Supplementary Table 5” if it is to remain in the supplement), however Table 5 does not give detailed information on the plasmid properties.

Response: We added DNA sequences of the CRISPR cassette to Supplementary Table 7 and provided corresponding descriptions in the section of results.

10. Line 67: While the SynIII marker appears to be absent by PCR (assuming a positive control would have worked), this fact alone cannot prove that the entire chromosome has been ablated. If the authors were to PCR for other markers distal to the SynIII site, then this may be able to lend credence to their claims, however it would still not be sufficient for complete proof of the chromosome’s absence. The claims outlined here require further support. While it appears this is addressed in Figure 2, the aforementioned claims can not be supported by the findings in Figure 1 alone.

Response: In addition to PCRTag analysis, whole-genome sequencing and Southern bolt analysis of the CRISPR-containing diploid strain showed that the entire target chromosome synIII was eliminated (Figure 2B; Figure 2C; Supplementary Figure 2; Supplementary Figure 3; Supplementary Figure 4).

11. Line 71-74: the genetics described is unclear here.

Response: We have rewritten the sentence in the revised manuscript.

12. Line 110: How sure can we be that it has duplicated? Or is there just only one viable chromosome available, and the other mostly cleaved? Did they count the total number of tetrads that came out of the experiment?

Response: A straightforward evidence of chromosome endoreduplication was illustrated by a fluorescence experiment through the process of chromosome drives (Figure 2A). Also, sequencing depth of duplicated chromosomes were used as the evidence for chromosome endoreduplication (Figure 2D). In addition, four spores were present for each tetrad in our experiment, indicating the duplications of target chromosomes in the driven diploid strains (if there is no duplication, only two spores are viable).

13. With the CRISPR components expressed episomally, could this be considered a drive? If they were to insert the CRISPR components into the genome, it could be considered a selfish chromosome.

Response: In our drive experiment of synX, chrXII, chrXIV and chrX, CRISPR components were inserted into the corresponding chromosomes (Figure 2; Figure 3). The details of CRISPR cassettes are shown in Supplementary Figure 5.

14. It would be a welcome addition for the authors to further elaborate on the practical application of such a system in the Discussion. Perhaps in industrial or agricultural applications, beyond mapping traits in a laboratory setting.

Response: Thank you for your suggestions. We have added the description of potential applications in industrial strain engineering in the discussion (line 287-294).

15. Line 230: I don't know if they can make an association between their system and homology-directed repair. I don't believe they sufficiently demonstrated that homing was occurring, nor that the chromosome was duplicating (however I do believe their system is successfully preventing the cleaved chromosome from being inherited – however this is not the same as duplication).

Response: Thank you for your suggestions. In drive systems like gene drive, it relies on the efficient homology-directed repair after DSBs. Chromosome drive system, however, relies on elimination of the target chromosome and following autonomous chromosome endoreduplication (not the traditional homology-directed repair). In our revised manuscript, a straightforward evidence of chromosome endoreduplication was illustrated by a fluorescence experiment through the process of chromosome drives (Figure 2A). Also, sequencing depth of duplicated chromosomes were also the evidence for chromosome endoreduplication (Figure 2D).

Minor concern:

1. Line 11: add “The” in front of target chromosome and “the” before desired chromosome for proper use of English grammar.

Response: We have fixed the problem in the revised manuscript.

2. Line 12: inclusion of the word “Whole” is considered inappropriate scientific language.

Response: We have fixed the problem in the revised manuscript.

3. Line 12: We understand this is likely an ESL manuscript. However there are significant grammatical mistakes, and improper use of scientific language throughout hindering reader comprehension. This review will therefore not seek to correcting for grammar, instead focusing the remainder of the review on Scientific merit and larger-scale errors under the assumption it will be proofed significantly prior to publication.

Response: The manuscript was edited for grammar, phrasing, and punctuation.

4. The sentence beginning in line 18 is a run-on. Please shorten for clarity.

Response: We have fixed the problem in the revised manuscript.

5. Figure 1A: denote exact distance from centromere for precision (ie, not <200bp)

Response: We have added the precise distance in Figure 1A.

6. Line 28: specify that it is “normal Mendelian inheritance” for accuracy and specificity.

Response: We have changed the word into “normal Mendelian inheritance”.

7. Line 33: reference 5 is incorrect

Response: We have changed the reference 5 with a new reference (Nat. Rev. Genet. 17, 146-159 (2016)).

8. Line 34: References 6-9 are exclusively mosquito gene drive papers, but the authors mention yeast and mammals as well. They should add or modify the references cited to reflect this.

Response: We have modified the references with more proper ones in the revised manuscript (References 9-16).

9. Line 40: It is not clear what the intended message of the sentence beginning with “It is still a challenge...” is. Please rephrase for clarity.

Response: We have fixed the problem in the revised manuscript.

10. Line 63: the gRNA target sequence should include the 5' and 3' annotations. And the PAM should be distinguished in some way. This should be done for all mentions of gRNA target sequence and primers throughout the text.

Response: We have added the 5' and 3' annotations to all the gRNA, primers and PAM sequences throughout the revised manuscript.

11. Line 72: figure should be capitalized.

Response: We have fixed the problem in the revised manuscript.

Reviewer #2 (Remarks to the Author):

In this study, the authors design a new type of gene drive system in yeast called a chromosomal drive. Cas9 and gRNA cut a site near the centromere of a chromosome, resulting in destruction of the chromosome. The drive chromosome, however, does not contain the target site, thus biasing inheritance toward the drive.

The toxin-antidote drive strategy was quite interesting. The paper will probably be impactful to the field and a more general audience. However, I do have some concerns that I believe must be addressed before the manuscript can be published. To give some perspective for this review, I specialize in gene drive but not in yeast genetics.

Major Points:

1. The entire manuscript requires a thorough edit for proper language. This will be necessary to publish the manuscript in an English-language journal. To help, here is a version of the abstract that has been minimally revised to correct several English errors:

“Self-propagating drive systems are capable of causing non-Mendelian inheritance and altering traits of populations. Here, we report a drive system called chromosome drive that eliminates a target chromosome and spreads a desired chromosome via CRISPR/Cas9. In our study, we found that a whole *Saccharomyces cerevisiae* chromosome can be eliminated efficiently through only one double strand break around the centromere using CRISPR/Cas9. As a proof-of-principle experiment of the CRISPR/Cas9 chromosome drives system, the synthetic yeast chromosome X was completely eliminated, and the counterpart wild-type chromosome X harboring a synthetic violetin pathway was driven throughout a population by sexual reproduction. We also demonstrate the use of chromosome drives to spread complex genetic traits in yeast. Megabase synthetic chromosome XII, heterogenous Y12 chromosome XIV, and SCRaMbLEd chromosome X were had biased transmission to progeny by chromosome drives in a population of *Saccharomyces cerevisiae* to spread a species identified ribosomal gene cluster from *Saccharomyces bayanus*, complex traits of thermotolerance, and unresolved traits of chromium resistance, respectively. Chromosome drives enables biased inheritance on a chromosomal scale, facilitating chromosome scale genetic mapping and extending applications of self-propagating drive.”

Response: Thank you for your suggestions. The entire manuscript was edited for grammar, phrasing, and punctuation to improve the flow and readability by Springer Nature Author Services.

2. Why does Figure 1B and 1C seem to imply 100% chromosome elimination, but then the experiments in Figure 3C indicate that elimination is usually 80-90% when the site is sufficiently close to the centromere?

Response: Figure 1B showed the analysis of all PCRTags for only one chromosome elimination strain yHX0295. In figure 1C, 78.78% (not 100%) of the diploid strains were capable of mating with tester α and growing on the corresponding test plate, which is comparative to the elimination efficiency of the site close to the centromere. Source data of colony numbers are listed in Supplementary Table 1.

3. Why weren't the yeast able to do end-joining repair for the Cas9-induced breaks. Is end-joining somehow inhibited near centromeres? If so, is this also true in other species? These matters should be included in the manuscript.

Response: Cas9-induced DSBs can be repaired by homology-directed repair (HDR) or nonhomologous end-joining (NHEJ). However, in yeast, frequency of DSB repairs via NHEJ was very low (~0.1%), comparing with homologous repair (Genetics 157, 579-589, 2001). In other organisms, the contribution of HDR and NHEJ to DSB repair is varied. These discussion were included in the revised manuscript (line 249-256).

4. The experimental details in Figure 2D are unclear, and I am unclear how to interpret the results of the spot

picture. In general, I understand that the chromosome drive increased in frequency over three rounds or generations of mating, but I'm unclear what the starting and ending frequencies of the drive were, which is quite important for this sort of experiment.

Response: We understand your concerns about Fig. 2D. We aimed to test the ability of the chromosome drive system through multiple rounds of mating and sporulation in this experiment. We have rewritten this description in the revised manuscript to make it more explicit (line 145-148).

5. Can the authors comment why the deletion efficiency is so different between the different strains in Supplemental Table 2? Is the cut rate lower, or did DNA repair (HDR or end-joining) occur? This might require a quick sequencing experiment, but this is critical for understanding what is happening in these yeasts.

Response: We randomly selected 40 off-target strains from the drive experiment of *synX*, *chrXII*, *chrXIV* and *chrX* and analyzed sequences around the targeted chromosomal region. We found that HDR happened in the CRISPR target region for the majority of the off-target strains in the experiments involving *synX*, *chrXIV* and *chrX*; however, the CRISPR target sequence remained intact for 9 of the 10 off-target strains in the drive experiment of *chrXII* (Supplementary Table 5). These results may be due to the low efficiency of CRISPR for *chrXII*, which contains a highly repetitive ribosomal gene cluster for the formation of a nucleolar structure, and there is also evidence that the efficiency of DSBs varies within different chromosomes. We added these in the discussion section (line 258-268).

6. Is there no recombination between chromosomes during yeast sexual reproduction? This could have an important effect on a chromosome drive and should be mentioned in the article if true (and discussed if not) to make it more accessible for non-specialist readers.

Response: There is recombination between chromosomes during yeast sexual reproduction. However, the process of elimination of a target chromosome and duplication of the homologous chromosome induced by a chromosome drive occurred before the process of meiosis, leading to loss of heterozygosity (LOH) in the diploid strains. It is possible that recombination occurs between the two identical chromosomes, however, there is no effect on the chromosome drive. Thank you for your suggestions. We have discussed this in the results section of *synIII* elimination (line 70-72).

Minor Points:

1. References 6 and 13 are the same. In general, in the introduction, it might also be useful to include several other CRISPR/Cas9 gene drives studies covering yeast and *Drosophila* (including a safeguarding study that used synthetic target sites).

Response: We have added more references that are related to CRISPR/Cas9 gene drives studies (including the safeguarding study) covering yeast and *Drosophila* in the revised manuscript (References 9-16).

2. In the first results subsection, it would be useful to make it explicit that the WT chromosome does not have the gRNA target site.

Response: We have added this description in the revised manuscript to make it more explicit.

3. Figure 1B is redundant. It can probably be eliminated, since all the information is in Supplementary Figure 1. However, why was the band for the wt-PCRTag not present in a few samples? Also, "Left-M" and "T" should be defined in the legend for Figure 1D.

Response: Thank you for your suggestions. We fixed these in the revised manuscript.

4. Supplemental Table 1 would be better if it had percentages, while still specifying the sample size of 120.

Response: We have added percentages in Supplementary Table 2.

5. The idea of targeting an important site with CRISPR with a desired site is immune to cleavage has been explored as a gene drive in *Drosophila*. They drives themselves are quite different, but they share the same "toxin-antidote" principle. Indeed, the authors in the discussion say: "To achieve the purpose of biased inheritance of desired chromosomes by chromosome drives system, the process of chromosome endoreduplication may not be critical since elimination of target chromosome in the zygotes will result in inviability of progenies without desired chromosome." This is the main method of previous CRISPR toxin-antidote drives and proposals. It might be worth including them in the introduction and/or discussion:

<https://www.nature.com/articles/s41467-020-14960-3>

<https://bmcbiol.biomedcentral.com/articles/10.1186/s12915-020-0761-2>
<https://www.biorxiv.org/content/10.1101/861435v2>

Response: Thank you for your suggestions. We have included it in the discussion section (line 279-281).

Reviewers' Comments:

Reviewer #2:

Remarks to the Author:

I am reviewer 2 from the initial set of reviews. The authors have overall made a good effort to improve the manuscript. With the following minor revisions, it should be suitable for publication in Nature Communications.

Regarding the points of my previous review:

1. The English language of the manuscript is substantially improved, but it still needs more work (I'm disappointed by the quality of the Nature editing service that the authors used - these issues should have caught by professional editors). Nevertheless, with the following additional revisions, it will be at an acceptable standard for publication. (In the following notation, the line listed is the starting line of the phrase to be corrected).

Line 29: "transposon" should be "transposons"

Line 44: "of *Saccharomyces cerevisiae* chromosome" should be "of a *Saccharomyces cerevisiae* chromosome".

Line 66: "exit" should be "exist".

Line 88: remove the word "And" (sentences cannot start with a conjunction)

Line 136: "were driven by the violacein pathway" should be "inherited the violacein pathway"

Line 137: "To verify chromosome elimination further," should be "To further verify chromosome elimination,".

Line 187: "To confirm whether the strain containing chrXIV of Y12 has a thermotolerance trait further, using chromosome transfer and chromosome elimination," should be "To further confirm whether the strain containing chrXIV of Y12 has a thermotolerance trait using chromosome transfer and elimination,"

Line 249: remove the word "cause".

Line 252: should "HR" be "HDR"?

Line 346: "1ml" should be "1 mL". There are also other instances in the method where "l" should be capitalized and there should be a space between the quantity and the measurement unit.

Line 566: "bolt" should be "blot"

Line 574: "deviations" should be "deviation"

Supplement Line 14: "bolt" should be "blot"

Supplement Line 50: "and three of them displayed in the figure" should be "and three of them are displayed in the figure"

Supplement Line 55: "Y12 behaved better thermotolerance compared to controls, indicating chrXIV of Y12 is one" should be "Y12 were more thermotolerant compared to controls, indicating chrXIV of Y12 is a"

Supplement Line 68: "BY4741 in diploid strain" should be "BY4741 in a diploid strain"

2-5. The revisions here are satisfactory.

6. I'm glad this experiment was done, since the paper is now much more complete. However, it should still be noted by the authors (probably in the discussion) that recombination could still occur in the ~20% of cases (highly variable depending on the strain and target) where the target chromosome is not eliminated. This could separate the target site from the desired chromosome to be eliminated. This would not a problem for single-generation experiments or controlled crosses over multiple generations. However, it would be an issue in the classic gene drive experiment in which a few drive individuals are released into a non-drive population and expected to take over the population over the course of several generations. In this case, the final population would not consist entirely of the desired chromosome unless the chromosome elimination efficiency was extremely high (enough to prevent surviving chromosomes, which could recombine). Including this discussion would improve the paper.

Minor points: All of these were suitably addressed.

I have a few additional minor points for improving the clarity of the revised manuscript.

1. Starting line 64, the authors write: "First, we designed a guide RNA (gRNA) targeting a specific sequence (5'- CATTATACGAAGTTATAAGTTGG-3') that is 135 bp from the centromere of synIII, which does not exist in chromosome III (chrIII) of BY4741. For biosafety and efficiency, a molecular containment strategy was applied¹⁰."

This sort of reverses the logical order of things. The fact that the synIII target site exists on there IS the containment strategy. I suggest uses the following wording:

"For biosafety and efficiency, a molecular containment strategy was applied¹⁰. We designed a guide RNA (gRNA) targeting a specific sequence (5'- CATTATACGAAGTTATAAGTTGG-3') that is 135 bp from the centromere of synIII, which does not exist in chromosome III (chrIII) of BY4741 or in wild-type yeast."

2. Within the supplement, the figures and tables should be labeled as "supplementary". They should also be labeled in the order they appear in the text (for example, supplementary figure 5 is the first one referred to, so it should be designed as #1 and come at the beginning of the supplemental section of the manuscript).

3. Supplement line 66 states that "Serial dilution assay analysis showed yHX0266 strains grew better than BY4741 under 40°C." However. I think the authors meant AT 40 degrees, rather than UNDER 40 degrees (though they also seem better at 37 degrees).

4. If possible, the panels in figure 3 should be arranged horizontally first and then vertically (so "B" should be where "D" is, "C" where "G" is, etc.).

5. Check the formatting of the authors in reference 5. In general, the formatting of many references has minor issues (the usual ones from Mendeley - too much capitalization of titles, lack of appropriate italics, occasional spacing issues, etc.).

6. The phrase "in an unbiased manner." on line 247 is a bit unclear, since the main objective is to bias the inheritance. Consider removing it.

7. Many of the semicolon breaks in the discussion should probably be completely separate sentences.

8. On line 279 if this occurs, it definitely employs the same principle, not just "may". The chromosome drive that the authors developed is certainly a form of CRISPR toxin-antidote gene drive in any case (with or without duplication).

9. On line 292, it's very unclear that chromosomal drive could be used in the treatment of diseases. End-joining is the dominant method in animals, and there is no sexual reproduction with somatic cells that would suffer from diseases. Consider removing this portion or elaborating on these factors.

10. I have a minor quibble with the other reviewer regarding the definition of a "gene drive". As noted in my original review, there are many toxin-antidote drives in existing (using both CRISPR and other methods) that work not by directly increasing the number of gene drive alleles, but by removing wild-type alleles from the population via various methods (thus indirectly increasing the frequency of gene drive alleles in the population). I think that the author's drive should therefore be called a "gene drive", though describing it as a "meiotic drive" is also certainly accurate. On another note, the experiment where the components are episomally inserted would be considered a "split drive".

Figure 2A seems to be pretty strong evidence of endoreduplication taking place, since I don't think there is a better mechanism to get all four members of the tetrad to have EGFP. I'm assuming that the image is typical of tetrads from this cross. Figure 2E also seems to support this, indicating that all four spores were viable.

Figure 2D based on sequencing depth could potentially be good evidence as well. My concern is that the authors may have just sequenced viable progeny, and of course, if a chromosome were lost, the individual would not be viable (thus preventing the "read depth" on ChrX from being reduced, even though at the tetrad stage, there would be half as many). Can the authors confirm that they were specifically sequencing tetrads here, which would ameliorate this issue? I wasn't clear on this based on the figure legend text, so they should probably revise the wording here.

Address to reviewer's comments

Reviewer #2 (Remarks to the Author):

I am reviewer 2 from the initial set of reviews. The authors have overall made a good effort to improve the manuscript. With the following minor revisions, it should be suitable for publication in Nature Communications.

Regarding the points of my previous review:

1. The English language of the manuscript is substantially improved, but it still needs more work (I'm disappointed by the quality of the Nature editing service that the authors used - these issues should have caught by professional editors). Nevertheless, with the following additional revisions, it will be at an acceptable standard for publication. (In the following notation, the line listed is the starting line of the phrase to be corrected).

Line 29: "transposon" should be "transposons"

Line 44: "of *Saccharomyces cerevisiae* chromosome" should be "of a *Saccharomyces cerevisiae* chromosome".

Line 66: "exit" should be "exist".

Line 88: remove the word "And" (sentences cannot start with a conjunction)

Line 136: "were driven by the violacein pathway" should be "inherited the violacein pathway"

Line 137: "To verify chromosome elimination further," should be "To further verify chromosome elimination,".

Line 187: "To confirm whether the strain containing chrXIV of Y12 has a thermotolerance trait further, using chromosome transfer and chromosome elimination," should be "To further confirm whether the strain containing chrXIV of Y12 has a thermotolerance trait using chromosome transfer and elimination,"

Line 249: remove the word "cause".

Line 252: should "HR" be "HDR"?

Line 346: "1ml" should be "1 mL". There are also other instances in the method where "l" should be capitalized and there should be a space between the quantity and the measurement unit.

Line 566: "bolt" should be "blot"

Line 574: "deviations" should be "deviation"

Supplement Line 14: "bolt" should be "blot"

Supplement Line 50: "and three of them displayed in the figure" should be "and three of them are displayed in the figure"

Supplement Line 55: "Y12 behaved better thermotolerance compared to controls, indicating chrXIV of Y12 is one" should be "Y12 were more thermotolerant compared to controls, indicating chrXIV of Y12 is a"

Supplement Line 68: "BY4741 in diploid strain" should be "BY4741 in a diploid strain"

Response: We have revised the manuscript according to the comments above.

2-5. The revisions here are satisfactory.

Response: Thanks.

6. I'm glad this experiment was done, since the paper is now much more complete. However, it should still be noted by the authors (probably in the discussion) that recombination could still occur in the ~20% of cases (highly variable depending on the strain and target) where the target chromosome is not eliminated. This could separate the target site from the desired chromosome to be eliminated. This would not be a problem for single-generation experiments or controlled crosses over multiple generations. However, it would be an issue in the classic gene drive experiment in which a few drive individuals are released into a non-drive population and expected to take over the population over the course of several generations. In this case, the final population would not consist entirely of the desired chromosome unless the chromosome elimination efficiency was extremely high (enough to prevent surviving chromosomes, which could recombine). Including this discussion would improve the paper.

Response: We have included this discussion in the revised manuscript.

Minor points: All of these were suitably addressed.

I have a few additional minor points for improving the clarity of the revised manuscript.

1. Starting line 64, the authors write: "First, we designed a guide RNA (gRNA) targeting a specific sequence (5'- CATTATACGAAGTTATAAGTTGG-3') that is 135 bp from the centromere of synIII, which does not exist in chromosome III (chrIII) of BY4741. For biosafety and efficiency, a molecular containment strategy was applied¹⁰."

This sort of reverses the logical order of things. The fact that the synIII target site exists on there IS the containment strategy. I suggest uses the following wording:

"For biosafety and efficiency, a molecular containment strategy was applied¹⁰. We designed a guide RNA (gRNA) targeting a specific sequence (5'- CATTATACGAAGTTATAAGTTGG-3') that is 135 bp from the centromere of synIII, which does not exist in chromosome III (chrIII) of BY4741 or in wild-type yeast."

Response: We have fixed the issue.

2. Within the supplement, the figures and tables should be labeled as "supplementary". They should also be labeled in the order they appear in the text (for example, supplementary figure 5 is the first one referred to, so it should be designed as #1 and come at the beginning of the supplemental section of the manuscript).

Response: We have labeled the figures and tables in the order.

3. Supplement line 66 states that "Serial dilution assay analysis showed yHX0266 strains grew better than BY4741 under 40°C." However, I think the authors meant AT 40 degrees, rather than UNDER 40 degrees (though they also seem better at 37 degrees).

Response: We have changed the word "under" with "at".

4. If possible, the panels in figure 3 should be arranged horizontally first and then vertically (so “B” should be where “D” is, “C” where “G” is, etc.).

Response: We have rearranged the figure3 horizontally.

5. Check the formatting of the authors in reference 5. In general, the formatting of many references has minor issues (the usual ones from Mendeley - too much capitalization of titles, lack of appropriate italics, occasional spacing issues, etc.).

Response: We have revised all references in the manuscript.

6. The phrase “in an unbiased manner.” on line 247 is a bit unclear, since the main objective is to bias the inheritance. Consider removing it.

Response: The phrase “in an unbiased manner.” has been removed.

7. Many of the semicolon breaks in the discussion should probably be completely separate sentences.

Response: The semicolon breaks in the discussion have been removed.

8. On line 279 if this occurs, it definitely employs the same principle, not just “may”. The chromosome drive that the authors developed is certainly a form of CRISPR toxin-antidote gene drive in any case (with or without duplication).

Response: We have fixed the issue.

9. On line 292, it's very unclear that chromosomal drive could be used in the treatment of diseases. End-joining is the dominant method in animals, and there is no sexual reproduction with somatic cells that would suffer from diseases. Consider removing this portion or elaborating on these factors.

Response: This portion has been moved.

10. I have a minor quibble with the other reviewer regarding the definition of a “gene drive”. As noted in my original review, there are many toxin-antidote drives in existing (using both CRISPOR and other methods) that work not by directly increasing the number of gene drive alleles, but by removing wild-type alleles from the population via various methods (thus indirectly increasing the frequency of gene drive alleles in the population). I think that the author’s drive should therefore be called a “gene drive”, though describing it as a “meiotic drive” is also certainly accurate. On another note, the experiment where the components are episomally inserted would be considered a “split drive”.

Response: Thanks for your comments.

Figure 2A seems to be pretty strong evidence of endoreduplication taking place, since I don't think there is a better mechanism to get all four members of the tetrad to have EGFP. I'm assuming that the image is typical of tetrads from this cross. Figure 2E also seems to support this, indicating that all four spores were viable.

Response: Thanks for your comments.

Figure 2D based on sequencing depth could potentially be good evidence as well. My concern is that the authors may have just sequenced viable progeny, and of course, if a chromosome were lost, the individual would not be viable (thus preventing the "read depth" on ChrX from being reduced, even though at the tetrad stage, there would be half as many). Can the authors confirm that they were specifically sequencing tetrads here, which would ameliorate this issue? I wasn't clear on this based on the figure legend text, so they should probably revise the wording here.

Response: We have added the corresponding description in the figure 2D legend text.